# The Effect of Temperatures on the Passivation Behavior of Q235 Steel in the Simulated Concrete Pore Solution

**DOI:** 10.3390/ma16020588

**Published:** 2023-01-07

**Authors:** Haosen Jiang, Zuquan Jin, Xiaoying Zhang, Lixing Qian, Zhaoliang Zhou

**Affiliations:** 1College of Civil Engineering, Qingdao University of Technology, Qingdao 266520, China; 2Engineering Research Center of Concrete Technology under Marine Environment, Ministry of Education, Qingdao 266520, China

**Keywords:** passive film, carbon steel, temperature, composition

## Abstract

Concrete, especially mass concrete, releases a large amount of heat during the hydration process, resulting in the passivation of reinforcement at high temperatures. However, the passivation study of reinforced concrete is mostly conducted at room temperature, and the influence of temperature on passive film behavior is not clear at present. The passivation film of reinforcing steel directly determines the corrosion resistance of reinforcing steel and affects the service life of reinforced concrete. Herein, the passivation of Q235 steel soaking in simulated concrete pore (SCP) solution at 20 °C, 40 °C, and 60 °C is explored. It is found that the passivation process is divided into two stages, with 24 h as the boundary; within 24 h the passivation was carried out rapidly, and the passive film is in a relatively stable state after 24 h. In addition, the higher the temperature, the faster the passivation. Moreover, under the condition of higher temperatures, more Fe^3+^ compounds are produced, and the semiconductor properties of passivated films are more stable. Based on experiments, the passivation mechanism affected by temperature was analyzed in detail.

## 1. Introduction

Reinforced concrete (RC) structures will suffer serious damage and cause huge economic losses due to the corrosion of steel reinforcement starting from the depassivation of the passive film on the surface of the steel bar [1,2,3,4]. The characteristics of the passive film are the response contributing to the anti-corrosion of steel bars and further affect the durability of RC structures [5]. The passive film is obviously very important for the actual service life of a project as the last barrier to the corrosion of reinforcing steel. Therefore, it is very important to deeply examine the performance of the passive film under different conditions.

In previous studies, the type of steel composition and the pH of the SCP solution have been proven to influence the anticorrosion of the passive film. Elsener et al. [6] studied the passive film composition and corrosion resistance of stainless steel in concrete and quantified C_crit_ in carbon steel for the first time. Li et al. [7] studied the passivation and corrosion behavior of carbon steel P355 in an SCP solution environment with different pH values and obtained the best pH value of passivation energy based on the test data. Based on this pH, the best passive film is obtained under the corresponding conditions. Mikimoto et al. [8] studied the passivation and corrosion behavior of high-carbon steel in a weakly alkaline environment and found that iron with high carbon content had better corrosion resistance after passivation.

In addition, some scholars have analyzed in detail the growth of the passive film in different solutions or the growth of alloy and stainless-steel passive films at different temperatures. Wu T et al. [9] investigated the effect of alternating temperature on the passivation characteristics of carbon steel in chlorine-free saturated Ca(OH)2 solutions and found that the passivation films formed in the variable temperature environment were more prone to rupture. Li Y et al. [10] investigated passivation film growth on carbon steel and its nanoscale characteristics at different passivation potentials, revealing the growth characteristics of passivation films on the nano scale. Feng et al. [11] studied the composition and corrosion resistance of passivated films with high H_2_S-CO_2_ at high temperatures. Wang et al. [12] studied the difference in the composition of passivated membranes at high temperatures in the presence of H_2_S, and a variation in the main composition of the passivation film was found. Wang et al. [13] explored the growth of the high-entropy alloy CoCrFeMoNi passive film at different temperatures and found that the passive film did not improve as we usually think with the increase of temperature, but on the contrary, although high temperature intensified the generation speed of the passive film, the passive film formed was not as good as that formed at normal temperature. The different simulated concrete solutions also have a large impact on the passivation film; for example, ordinary hole solutions, hole solutions containing red mud, hole solutions containing sulfide, and hole solutions with their own chloride ions all have a large impact on the passivation film formed [14,15,16,17]. These results indicated that the temperature plays an important role in the passivation behavior of the alloy stainless steel. However, up to now, as the main materials used in buildings, the research on the passivation behavior of concrete reinforcement at different temperatures is still relatively lacking. 

In this paper, an electro-chemical workstation was used to continuously test OCP and EIS on carbon steel passivated at different temperatures to obtain information on the growth of the passive film during the passivation process. The Mott–Schottky (MS) curves of the samples were tested before and after passivation to comprehensively evaluate the performance of the passive film. Scanning electron microscopy (SEM) and X-ray photoelectron spectroscopy (XPS) were used to observe the microstructure and composition of the passive film. Finally, the mechanism of passivation affected by temperature was analyzed comprehensively.

## 2. Experiments

### 2.1. Materials

The steel block used in the experiment is precision-cut directly from Q235 carbon steel and then slightly polished and stored in kerosene. The Q235 size is 10 mm × 10 mm × 5 mm. Other chemical compositions are shown in Table 1. Super-clear epoxy AB glue is used for epoxy. The mold is a silica gel mold with a diameter of 2 cm and a height of 1.8 cm.

### 2.2. Sample Preparation

The Q235 steel block in kerosene was removed and placed in a beaker with anhydrous ethanol and ultrasonically cleaned for 10 min. It was then taken out; the anhydrous ethanol was dried with a hair dryer, and wire welding was performed immediately. After welding, the sample was put into a silicone mold and poured into epoxy resin. The electrodes were treated and polished with 220#, 400#, 800#, 1500#, and 2000# sandpaper. Finally, water was removed from the electrode surface, and the electrode was placed in a vacuum-drying tray to preserve it. For the samples under the microscope, after ultrasonic cleaning, they were directly put into the configured alkaline solution for passivation. As shown in Figure 1 below, they are the two samples needed for this experiment. The electrodes in figure a were used for electrochemical testing and the samples in Figure 1b were used for microscopic testing

### 2.3. Environment

The conditions are 20 °C, 30 °C, 40 °C, 50 °C, and 60 °C. The pH is 13.5. The SCP was prepared using saturated calcium hydroxide (Ca(OH)_2_), potassium hydroxide(KOH), and sodium hydroxide (NaOH) solutions. Because the solubility of calcium hydroxide decreases with increasing temperature, after the solution is configured, it should be placed in the oven at the corresponding temperature, heated for 24 h, and then be made fully stationary before removing the supernatant.

### 2.4. Electrochemical Test

The electrode was passivated in the reducing solution at 20 °C, 40 °C, and 60 °C. An electrochemical workstation (5000 E) from Gamry was used to continuously collect the corrosion potential and impedance (100 kHz–0.1 Hz) of the Q235 electrode in the passivation process (during the data collection process, the electrolytic cell was put into a water bath to keep the temperature constant during the passivation process), and the MS curve of the electrode before and after passivation was tested; the performance of the passive film was comprehensively evaluated, and its erosion-carrying ability was judged by OCP and EIS.

### 2.5. Microscopic Morphology Observation

We removed the Q235 carbon steel passivated at different temperatures, immediately put it into absolute ethanol to clean the passivated solution, and then used a hair dryer to dry the sample with cold air, keeping it in a vacuum-drying dish to prevent corrosion in the air. Note that the passive film is easy to rupture when exposed directly to air, so a complete test surface should be retained by picking it up with tweezers. Finally, scanning electron microscopy was used to observe the microstructure of the passive film.

### 2.6. Composition of Passive Film

X-ray photoelectron spectroscopy (XPS) was used for analyzing the passive film developed on the surface of the steel bar in the SCP solution at different temperatures. Spectra of Fe and O elements were recorded. Argon ion sputtering was used to obtain the chemical composition of the passive film at depths of 0 nm, 2 nm, 4 nm, and 8 nm.

## 3. Results and Discussion

### 3.1. Corrosion Potential Curve

The OCP was used to continuously collect the potential changes of Q235 soaking in SCP at 20 °C, 40 °C, and 60 °C for 240 h with a three-electrode system, where a platinum sheet electrode, mercury oxide electrode, and prepared Q235 electrode served as a counter electrode, reference electrode, and work electrode, respectively. It can be seen in Figure 2 that the slope change of the OCP curve is divided into two stages. In the first 24 h (stage I), the corrosion potential of Q235 at 20 °C, 40 °C, and 60 °C increases rapidly from −380 mV to about −150 mV. After 24 h, all the corrosion potential curves steadily rise. Differently, 60 °C finally reached −120 mV, 40 °C followed, reaching about −130 mV, and 20 °C was the smallest at 145 mV. The higher the temperature, the greater the positive shift of the corrosion potential, indicating that the passivation effect is enhanced with an increase in temperature.

### 3.2. Characteristics of Passive Film

#### 3.2.1. Characteristics of Passive Film on the Surface

As shown in Figure 3 and Table 2, the XPS test was carried out on the surface of the passive film. There are four peaks formed by Fe, including Fe2p 3/2 and 1/2, in which Fe^0^ is about 706.7 eV and 719.8 eV, Fe^2+^ is around 709.7 ± 0.2 eV, and Fe^3+^ -O is approximately 710.4 ± 0.2 eV and 723.9 ± 0.2 eV. The Fe^3 +^ -OH values were 711.5 ± 0.2 eV and 724.3 ± 0.2 eV. Furthermore, it can be found that with the increase of temperature, the content of Fe^0^ decreased significantly; the values at 20 °C and 40 °C were similar, while the content of 60 °C iron decreased significantly, being only 9.05%. The Fe^2+^ content dropped significantly at 40 °C, reaching 6.52%, and with the temperature further rising to 60 °C, the Fe^2+^ content decreased again. Fe^3+^, on the other hand, increases significantly with increasing temperature, reaching 85.43% at 60 °C. Among them, Fe^2+^ mainly comes from Fe_3_O_4_, while Fe^3+^ comes from Fe_2_O_3_ and FeOOH. Obviously, the increase in temperature produces less Fe_3_O_4_ that is easy to decompose and produces more stable ferric compounds. This is similar to the findings of previous studies [18,19].

From the perspective of oxygen, the hydroxyl compounds decreased significantly with the increase of temperature, and the water on the surface of the passivation membrane was also relatively reduced, while oxygen ions increased significantly, reaching 43.10%. Overall, as far as the passive film surface is concerned, for the rise of temperature, the passive film contains more Fe_2_O_3_, and the passive film is more corrosion-resistant [20].

#### 3.2.2. Characteristics of Passive Film at Different Depths

As shown in Figure 4, as the temperature increases, it is obvious that the peak of iron elements is smaller, and this law is also followed in the depth direction. Especially at 8 nm, the peak strength of iron elements at 20 °C is very obvious, indicating that with the increase of depth, it is closer to the iron matrix, while the passive film formed at a higher temperature over peak strength, indicating that the increase of temperature makes the formed passive film thicker. From the point of view of oxygen, with the increase of depth, the oxygen in the passivation film produced at higher temperatures also confirms from the side that the passivation film is thicker with increasing temperature. In addition, with the increase of depth, the peak of OH^–^ also gradually increases, especially at higher temperatures, and the peak strength is larger. When the two elements are combined, the inner layer grows first and extends outward because the passive film is formed in layers. So, the outer layer has more Fe_3_O_4_, and the inner layer has more Fe_2_O_3_ and FeOOH; this is because the first ferro-bivalent compound, Fe_3_O_4_, is converted into Fe_2_O_3_ and FeOOH.

### 3.3. EIS Evolution of Q235 Steel during Passivation

To further study the formation process of the passive film on the surface of Q235 carbon steel, electrochemical impedance tests were performed. Figure 5, Figure 6 and Figure 7 show the diagram of the selected equivalent circuit, Nyquist, and Bode diagrams, respectively. The fitting results are listed in Table 3.

The EIS data were fitted, and a reasonable equivalent circuit was selected according to the Nyquist diagram and relevant references [9], as shown in Figure 6. *Rs* is the internal resistance, that is, the resistance of the electrolyte and the working electrode itself. *R_f_* is the Faraday resistance, including electron transfer resistance and substance exchange resistance, and *Q_f_* is the double electric layer potential, changing the electron configuration at the interface. *R* and *Q_dl_* are the resistors and capacitors of the passive film, respectively.

Generally, it is believed that the high-frequency region of the electrochemical impedance spectrum of the rebar is controlled by the charge transfer process because the frequency changes too fast to allow for material transfer. It reflects the information on the charge transfer resistance of the rebar and the capacitance of the double layer, and the radius of the capacitive reactance arc represents the charge transfer resistance. The low-frequency region is controlled by material transfer, which reflects the signal feedback of the resistance and capacitance of the passive film. The radius of the capacitive arc reflects the resistance of the passive film on the surface of the steel bar [21,22].

In the initial stage of passivation, the impedance of the passive film at high frequency increases rapidly, and the higher the temperature is, the greater the growth rate is. In the high-frequency part, the change is no longer obvious after 12 h, while the low-frequency part does not remain stable until 24 h later. In the middle and late stages of passivation, the impedance of the high-frequency part is more stable when the temperature is lower, and the impedance at 60 °C relatively fluctuates greatly. The impedance of the low-frequency part is consistent with that of the high-frequency part. The impedance of the low-frequency part decreases obviously at 60 °C and a little at 40 °C. The passive film is relatively stable at 20 °C. These changes in the impedance of the passive film result from the rapid passivation reaction. With the increase in temperature, the molecular reaction force increases, and the formation of the passive film is accelerated, which is the reason why the impedance increases rapidly in the early stage, and the higher the temperature, the greater the impedance. However, with the increase of the passivation time, the impedance of the passive film decreases at a higher temperature because Fe_3_O_4_ in the outer layer of the passive film is not stable after a long time of high-temperature soaking. One part is converted into more stable Fe_2_O_3_ and relatively stable FeOOH, and the other two parts are dissolved into the SCP solution.

Overall, the passive film generated at 60 °C has the fastest growth. In the initial stage, from the point of view of reaction kinetics, it has almost three times the impedance of the passive film at 20 °C and 40 °C. However, at 48 h of passivation, the impedance of the passive film continues to grow at 20 °C, taking the leading position, while the impedance of the passive film at 40 °C and 60 °C decreases slightly. This is because the passive film growing in the outer layer is not stained in an environment with higher-than-normal temperature, resulting in little dissolution. This is also why the corrosion potential has a slightly negative shift in the early stage.

Table 3 shows the fitting resistance and electrochemical resistance parameters of Q235 steel after passivation at different temperatures, and the fitting error ranges from 10^−3^ left and right, indicating that there is good agreement between the measured data and the fitting data. It can be seen from the table that under the same temperature strip, with the increase of soaking time, the pure film resistance value Rf on the surface of Q235 steel gradually rises higher, especially its passivated membrane impedance in the first 24 h of initial soaking. The increasing amplitude is especially obvious, and in the same soaking time, when the temperature of the solution rises, the impedance of the passive film is measured, and there is a trend of gradual increase. The above results indicate that as the temperature of the solution rises, the growth rate of the pure film increases, resulting in higher corrosion resistance of the passive film in the same passivation time. This is because, as the temperature of the solution rises to result in a dull film of ferrous 2, the oxygens gradually change into trivalent oxygens and hydroxyl oxides; thus, the corrosion resistance of the passive film is enhanced [23].

The resistance at 40 °C and 60 °C of the passive film deteriorates slightly in the middle and late stages, which is consistent with the Bode and Nyquist plots of the later impedance changes. In particular, in the 60 °C Bode plot at the last moment of passivation, the decrease in impedance can be seen. Because the electrode is taken out of the oven at a constant temperature during the test, and during the test for about 1 h, a temperature change occurs, the alternating temperature causes microcracks on the surface of the passive film, so the impedance decreases [24]. During the entire passivation reaction, the internal resistance is small, and the temperature does not cause too much interference. The charge transfer resistor is relatively stable and is not greatly affected by temperature.

### 3.4. Reaction Rate

The following equation is the Arrhenius formula, where K is the rate constant, *Ea* is the activation energy, *R* is the molar gas constant, *T* is the thermodynamic temperature, and C is the constant.
(1)$$\mathrm{InK}=\frac{-Ea}{RT}+C$$

According to the experimental conditions and converting the temperature, K takes the reciprocal 1/R_ct_ of the electron transfer impedance when the passive film starts the reaction, and the value range and results are shown in Table 4 below.

With the increase in temperature, the activation energy of the initial passivation reaction gradually decreases, and the difference is about 1.7 KJ/mol. The percentage of activated molecules is also higher and the reaction rate is faster, which is why the positive shift in corrosion potential with increasing temperature is more pronounced in the primary passivation phase.

### 3.5. Micro-Structure

As shown in Figure 8, the surface morphology of the passive film varies greatly at different temperatures. Under magnifications of 500, 3000, and 5000 times, we can see that the surface compactness of the passive film decreases gradually with an increase in temperature. The passive film surface at 60 °C is densest, and it is least dense at 20 °C, which can well prove the dissolution of the passive film surface at 40 °C and 60 °C, which is also the reason why the impedance at 40 °C and 60 °C of the passive film decreases in the middle and late period. The surface of the 20°C passive film is densely covered by the growth of Fe_3_O_4_. As the temperature increases, the corresponding reaction products gradually decrease. Although it is not very dense, it increases the impedance value of the passive film. On the whole, the passive film forming at 60 °C is better, followed by that at 40 °C, while the worst film forms at room temperature.

### 3.6. Mott–Schottky Curve

Under natural passivation, the condition is discussed through a capacitance experiment. As shown in Figure 9, the MS curve shows two regions with an increase of applied potential at different temperatures. In the application potential range of −0.8–0.4, the capacitance value decreases with the increase of applied potential value, and the curve increases roughly linearly, indicating that the passive film has N-type semiconductor characteristics; it has extra electrons in the band gap. However, in the potential range of 0.4–0.7, the capacitance increases, and the curve decreases roughly linearly, indicating that the passive film has P-type semiconductor characteristics, and its outermost shell is not fully full. This can be attributed to the double-layer structure of the passive film. At low potentials, the outer donor level is mainly controlled by divalent iron oxide, but at high potentials, the deep donor level is controlled by ferric iron oxide. Thus, there will be an extra electron in the outermost orbital that is full of electrons, or there will be an electron in the outermost orbital that cannot be bonded to for lack of another electron. After the passive film is stabilized, it can be seen that the MS curves at 40 °C and 60 °C coincide, which indicates that the semiconductor properties of their corresponding passive films are the same. At the same time, in the P-N junction formed by the passive film, the N-type difference is large, and the N-type semiconductor can be used to comment on the overall performance of the passive film. P-N junctions have formed.

The flat band potential of the passive film at different temperatures is given, and the N-type semiconductors formed by the passive film at different temperatures are significantly different, both during the initial and final stages of passive film formation. The smallest slope is for the passive film at 60 °C, indicating that the N-type semiconductor carrier concentration it forms is lower, there are fewer vacancies in the passive film, the film is denser, and the formation is better. While the film at 40 °C is second, the passive film at room temperature is the worst, which is consistent with the SEM diagram. Among them, the above figure marks the flat band potential of the passive film N-type semiconductor at different temperatures, and there is no large change in the flat band potential of the film at 60 °C before and after the initial period, indicating that the formation of the passive film is faster. The flat band potential of the passive film at room temperature and 40 °C changes greatly before and after, which is related to the formation rate of the passive film at those temperatures.

### 3.7. Passivation Mechanism

As shown in Figure 10, the passivation reaction is a very complex and long process. In combination with the test, we derived the process of the passivation reaction. Regarding the passive film’s continuous growth in the passivation process, according to the experimental results, we can see that as the temperature increases, the passive film is thicker, the passivation is faster, and at the same depth, the higher-temperature passive film has more trivalent iron. According to the Arrhenius formula, the passive film’s material composition at different temperatures is roughly the same, and the high temperature only intensifies the speed of the reaction. That is to say, the passivation effect that can be completed in 10 h at 20 °C may be completed in 5 h at 40 °C. Based on this, when the final passivation is complete, more ferric iron produced at high temperature is converted from divalent iron, so Fe_3_O_4_ is first produced at the beginning of passivation and then converted into Fe_2_O_3_ and FeOOH; As a result, the outermost layer of the passivation film has more divalent iron, which gradually decreases with increasing depth, while the trivalent iron is increasing.

## 4. Conclusions

In this study, the passivation behavior of Q235 carbon steel at different temperatures of 20 °C, 40 °C, and 60 °C was studied. The specific conclusions are as follows:(1)The passivation process was divided into a rapid passivation phase before 24 h and a stable passivation phase after 24 h. Among them, the early passivation rate was faster at 40 °C and 60 °C, indicating that the increase in temperature accelerated the formation of the passive film.(2)Under different temperature conditions, the passive film products are not the same, and more Fe_2_O_3_ is produced with an increase in temperature—which is the main reason for the temperature increases—and the passive film is better and thicker.(3)After passivation, the semiconductor properties of the passive films formed are the same. At a temperature of 20 °C, the capacitance of the passive film after passivation is significantly smaller than that at 40 and 60 °C. As the temperature increases, better passivated films form.

## Figures and Tables

**Figure 1 materials-16-00588-f001:**
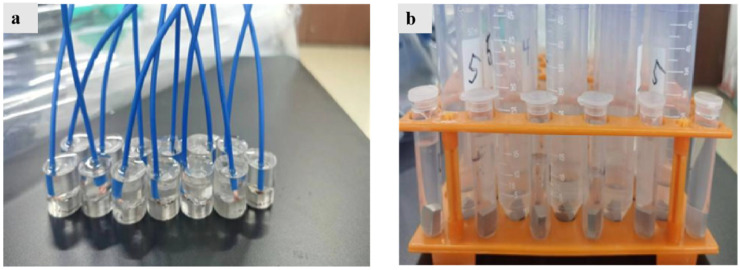
Test samples: (**a**) the electrode samples and (**b**) the micro samples.

**Figure 2 materials-16-00588-f002:**
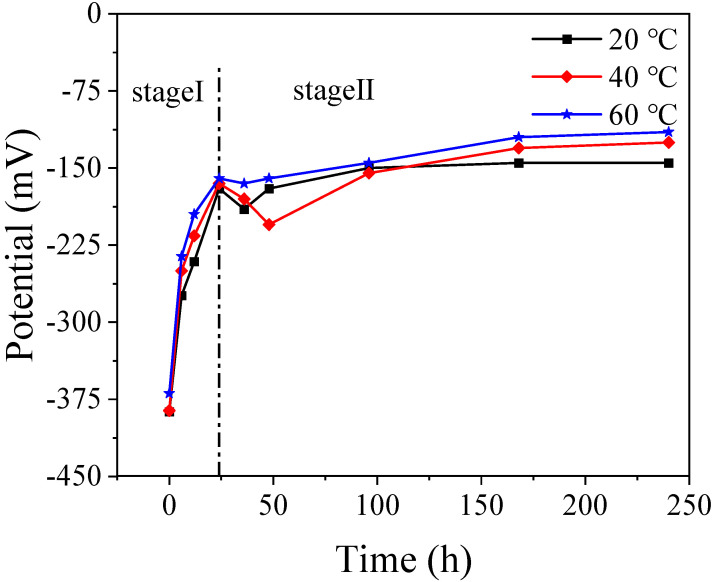
The OCP evolution of Q235 steel after passivation in SCP at various temperatures for a different time.

**Figure 3 materials-16-00588-f003:**
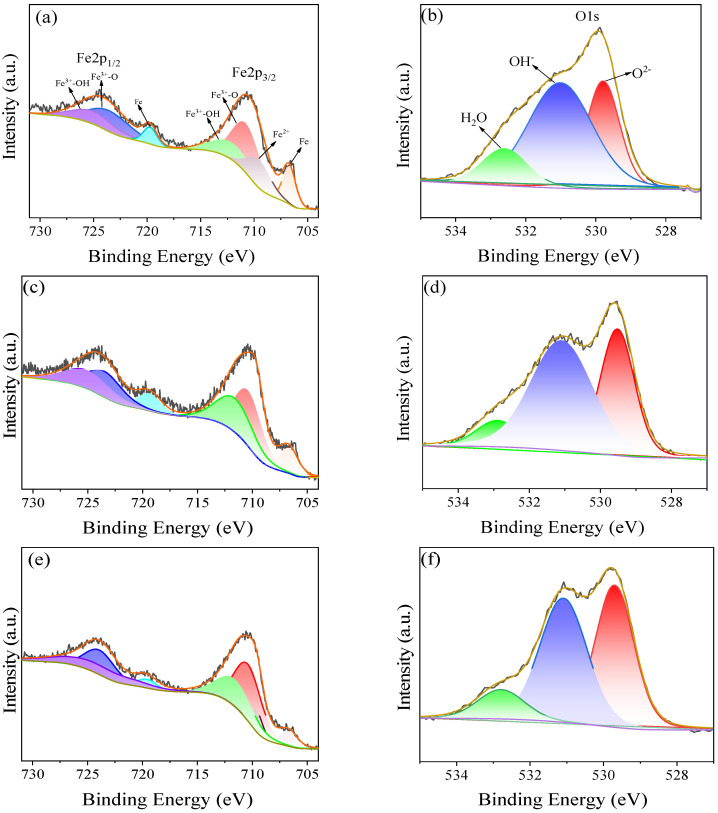
Characteristics of passive film on the surface at different temperatures: (**a**,**c**,**e**) are the XPS diagram of iron, and (**b**,**d**,**f**) are the XPS diagram of oxygen.

**Figure 4 materials-16-00588-f004:**
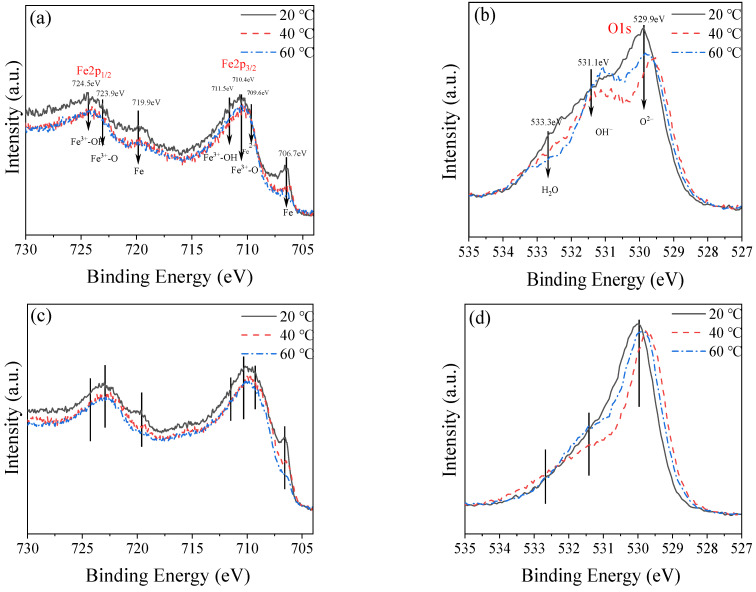
Characteristics of passive film at different depths: (**a**,**c**,**e**,**g**) are the XPS diagram of iron, and (**b**,**d**,**f**,**h**) are the XPS diagram of oxygen.

**Figure 5 materials-16-00588-f005:**
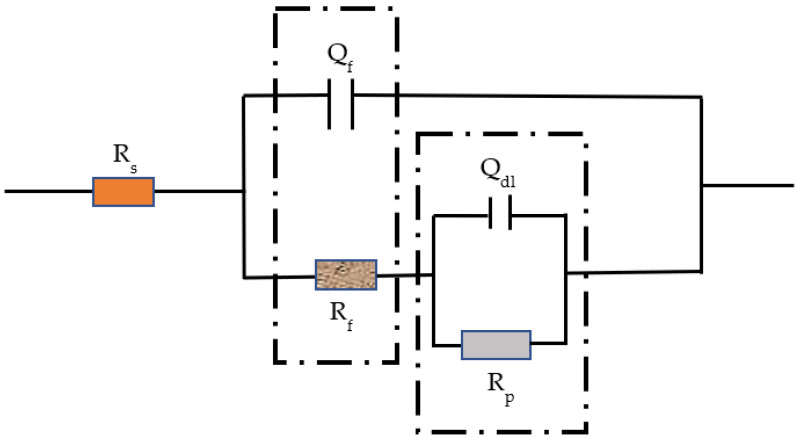
Equivalent circuit diagram.

**Figure 6 materials-16-00588-f006:**
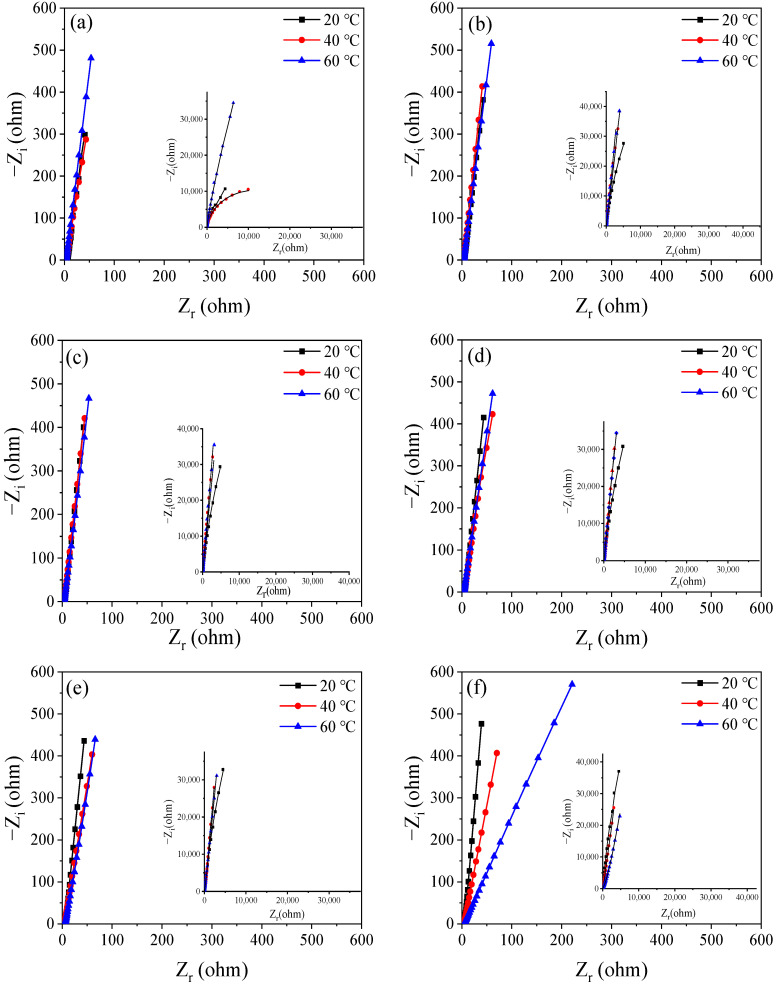
Nyquist plots of Q235 steel passivated for (**a**) 0.5 h, (**b**) 6 h, (**c**) 12 h, (**d**) 24 h, (**e**) 48 h, and (**f**) 240 h.

**Figure 7 materials-16-00588-f007:**
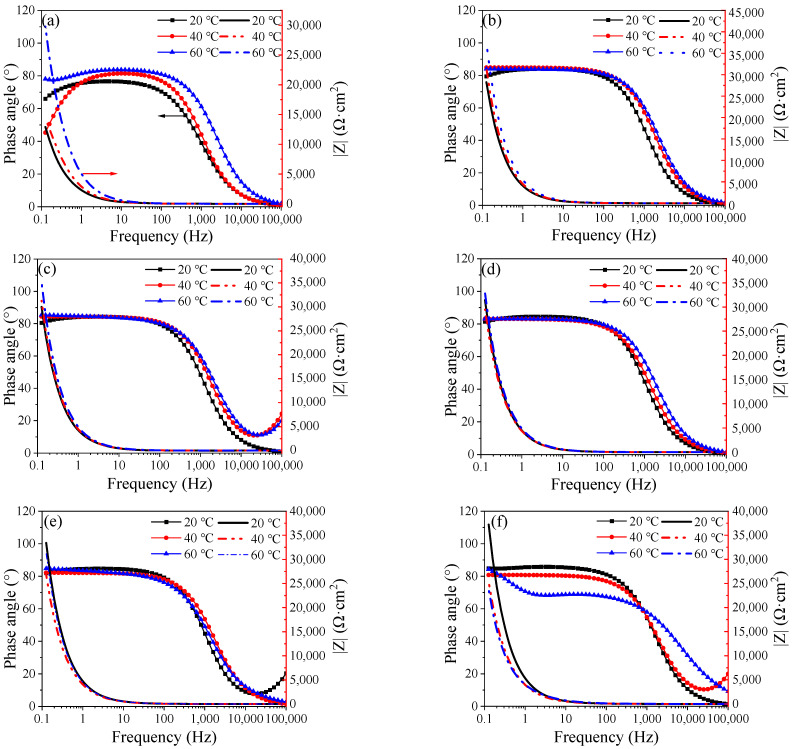
Bode plots of change with the duration of passivation time: (**a**) 0.5 h, (**b**) 6 h. (**c**) 12 h, (**d**) 24 h, (**e**) 48 h, and (**f**) 240 h.

**Figure 8 materials-16-00588-f008:**
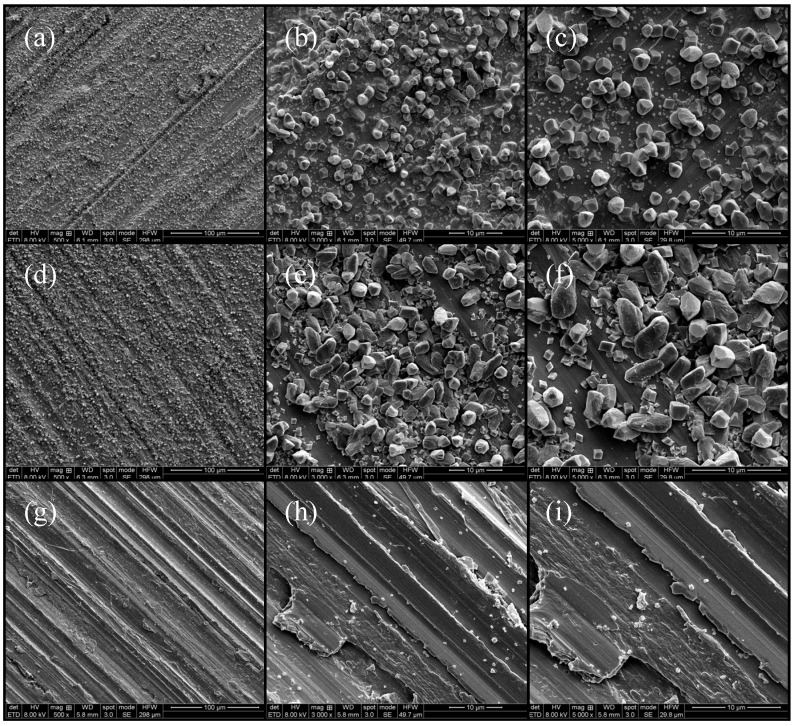
SEM diagrams of passive film at different temperatures: (**a**–**c**) are passive film at 20 °C, (**d**–**f**) are passive film at 40 °C, and (**g**–**i**) are passive film at 60 °C.

**Figure 9 materials-16-00588-f009:**
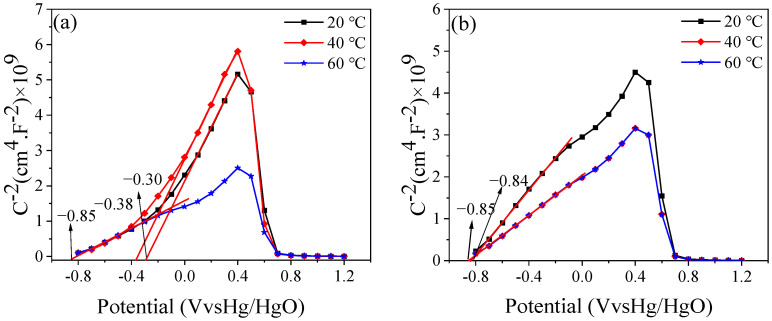
MS curve of Q235 steel: (**a**) 0.5 h and (**b**) 240 h.

**Figure 10 materials-16-00588-f010:**
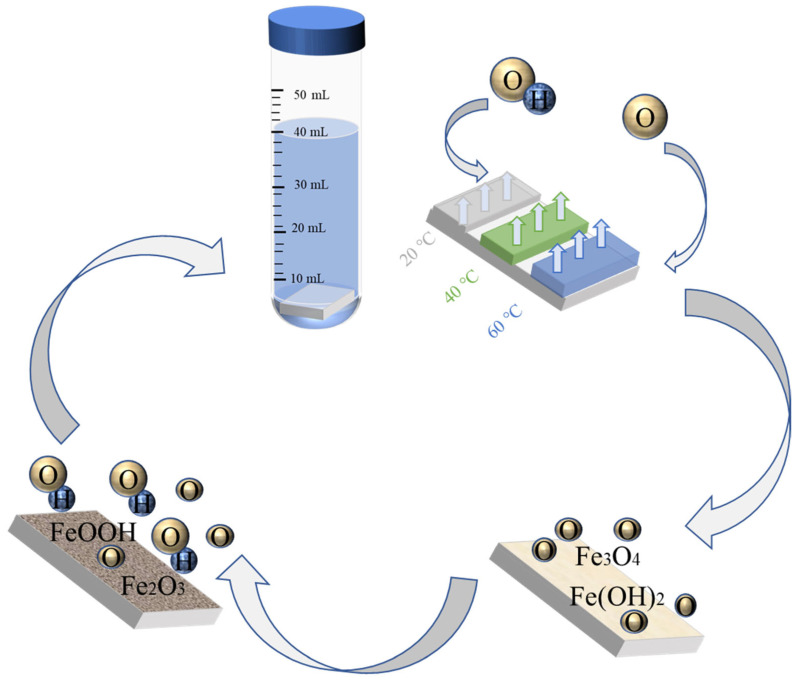
Schematic diagram of the passivation mechanism of Q235 steel.

**Table 1 materials-16-00588-t001:** Compositions of Q235 steel.

C	Mn	Si	S	P
0.20%	1.4%	0.35%	0.045%	0.045%

**Table 2 materials-16-00588-t002:** Peak area percentage.

	20 °C	40 °C	60 °C
Fe^0^	1.27	1.31	0.82
Fe^2+^	1.15	0.57	0.50
Fe^3+^	6.15	6.84	7.72
H_2_O	6.21	3.74	4.44
OH^−^	24.75	22.64	18.87
O^2−^	13.11	14.23	17.66

**Table 3 materials-16-00588-t003:** EIS fitting parameters of Q235 carbon steel at different temperatures and passivation times.

T (°C)	Time (h)	*R_s_* (Ω·cm^2^)	*Q_f_* (μF^−1^ cm^2^ s^n^)	n	*R_ct_* (Ω·cm^2^)	*Q_p_* (μF^−1^ cm^2^ s^n^)	n	*R_p_* × 10^5^ (Ω·cm^2^)	*χ^2^* (×10^−3^)
20 °C	0.5	5.844	0.887	0.86	6.21	0.36	0.79	0.77	2.92
	6	5.166	0.475	0.94	3.13	0.21	0.83	1.50	5.19
	12	5.198	0.144	1.00	5.18	0.41	0.94	3.69	9.43
	24	5.468	0.110	1.00	6.18	0.39	0.95	4.59	7.12
	36	5.876	0.116	1.00	6.50	0.38	0.94	5.39	2.20
	48	5.162	0.108	0.99	6.40	0.37	0.95	6.16	1.23
	96	5.222	0.339	0.95	2.44	0.72	0.78	8.09	7.16
	168	5.767	0.334	0.95	2.13	0.97	0.95	8.45	2.85
	240	4.657	0.331	0.95	2.50	1.11	0.88	8.75	5.93
40 °C	0.5	4.546	0.609	0.93	1.64	0.52	0.63	1.03	2.56
	6	3.624	0.397	0.94	3.97	0.14	0.74	3.11	1.46
	12	3.762	0.401	0.94	8.99	0.48	0.82	5.86	1.17
	24	5.276	0.421	0.92	5.22	0.10	0.86	6.42	1.01
	36	4.491	0.444	0.92	6.24	0.11	0.84	5.46	6.40
	48	4.302	0.408	0.99	6.41	0.37	0.85	6.16	1.23
	96	3.693	0.470	0.91	6.13	0.86	0.86	5.60	2.31
	168	4.107	0.469	0.90	5.09	0.10	0.85	3.53	2.47
	240	4.275	0.489	0.90	5.57	0.37	0.83	3.26	7.99
60 °C	0.5	3.474	0.353	0.93	1.24	0.98	0.86	2.62	1.96
	6	3.965	0.333	0.93	2.13	0.56	0.82	5.36	1.89
	12	3.682	0.356	1.00	3.72	0.34	0.93	6.49	2.59
	24	4.363	0.377	0.92	7.79	0.83	0.87	7.69	1.08
	36	4.098	0.415	0.92	3.52	0.93	0.82	7.46	1.21
	48	5.710	0.415	0.91	1.64	0.36	0.80	7.32	9.79
	96	4.598	0.470	0.90	1.78	0.47	0.87	6.21	5.65
	168	4.544	0.503	0.94	1.03	0.81	0.86	5.48	2.31
	240	4.394	0.566	0.97	1.38	0.97	0.73	5.32	1.09

**Table 4 materials-16-00588-t004:** Calculated values and results.

T (°C)	lnk	1/T (1/K × 10^−3^)	Ea. (KJ/mol)
20	−1.8262	3.41	28.35
40	−0.4947	3.19	26.52
60	−0.2151	3.00	24.94

## Data Availability

This study will continue to explore the refinement, Data sharing is not applicable to this article.

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
