# Peer review of "The Effect of Temperatures on the Passivation Behavior of Q235 Steel in the Simulated Concrete Pore Solution"

_materials, 2023, doi:10.3390/ma16020588_

Round 1

Reviewer 1 Report

Dear authors

The manuscript is well presented and contains interesting results about the passivation behavior of Q235 carbon steel at different temperatures of 20°C, 40°C, and 60°C.

I recomend the acceptance of the paper as it is

Reviewer 2 Report

I found this study quite interesting and practically useful. The study of steel passivation under different temperature regimes is quite an important aspect of future research in the advancements of steel-reinforced mass concrete structures. The findings of this study are quite genuine. However, I think minor revision is required before the finalization of this work for acceptance:

1)     The significance of studying the passive behavior of steel should be briefly mentioned in the abstract and introduction.

2)     What are the possible solutions to control the passive behavior of steel? The role of admixture is quite significant in controlling the heat of hydration.

3)     First, para-introduction, do not use too many references for a simple piece of information. Rather focus on explaining each study knowing that each study is distinct.

4)     The reference number/citation must be mentioned immediately after the names of the authors for instance., Feng et al., [3].

5)     The problem statement is quite generic and must be differentiated from the existing research.

6)     Page 3-Line1; the temperatures are not written correctly according to standard.

7)     In-text citations of figures must be revised. You cannot write just. As shown in Fig. 1 and end the sentence that way.

8)     The results and discussion section are satisfactory but the results still need validation.

9)     Conclusions must be enlisted in short and crispy bullet points.

Reviewer 3 Report

The paper “The effect of temperatures on the passivation behavior of Q235 steel in the simulated concrete pore solution” is devoted to characterizing the passivating films after the different treatment conditions. The structural and surface properties of obtained films are examined. Generally paper seemed to be suitable for publication in Materials after the revision.

1)         I suggest not showing table.2 as it contains nonsense information (violating the percentage definition: the combined values are higher than 100).

2)         Wide XPS scans also should be presented to prove there are only iron and oxygen photoelectron and Auger lines. Please do not omit C1s, N1s, and manganese core levels (despite their presumable presence).

3)         Authors indicate the presence of different oxidation states of Fe by decomposition of Fe 2p line. The numbers of decomposed oxidation states differ for 2p 3/2 and 2p ½. That is why the quantitative analyses were done only for the O1S line. I suggest that the Fe3p line is also informative as electrons’ escape depth for Fe 3p is essentially higher than that one for Fe2p. Their analyses may prove the distributions of oxygen bonds near the surface and more close to the bulk. Results from fig. 4, thus, could be reinterpreted. Please also provide a deconvolution method description to understand the nature of curvy baselines for the iron core levels.

4)         References [4-5] should be cited as a single position.

Reviewer 4 Report

In this paper, the passivation of Q235 steel by immersion in simulated concrete pore (SCP) solution at 20°C, 40°C and 60°C was investigated. For this, a unique experimental study was carried out in terms of materials, preparation of samples, electrochemical testing and microscopic morphology observation. The organization of the research carried out in this paper was found to be very successful and the authors are congratulated. I have brought the following minor revisions to the attention of the authors as a contribution to the paper only.

1-- For the notation and abbreviations used/reported in the various main-titles and/or sub-titles of the paper; it is suggested that an additional sub-heading be created for this, where deemed appropriate by the authors, in order to follow the focus of the paper in a healthier way and to create integrity.

2-- The grammatical and punctuation deficiencies noticed in the first paragraph of the "Samples preparation" sub-title and the incomplete sentences should be checked by the authors for a healthier understanding. In other words, it is meant to confirm whether the chronological operations made in the preparation of the samples were carried out by the authors and to combine the last two sentences of the paragraph.

Since a similar situation is noticed in the next paragraph of this sub-heading, it is recommended that this note be checked throughout the paper. In addition, the names of Sodium hydroxide (NaOH) Potassium hydroxide (KOH) solutions that were misreported in the paper should also be corrected.

3-- For the sub-title "OCP curve during passivation of Q235 electrode", considering Fig. 2, the 45 mV measured for Potential at 20°C needs to be verified by the authors.

4-- Figure 6 and Figure 7 under the sub-title of "EIS evolution of Q235 steel during passivation" are considered to need some further discussion, as it is accepted by the authors that the passivation process is limited to 24 hours, as stated in the paper.

5-- The Arrhenius formula used in the "Reaction rate" sub-title needs to be numbered and the word "The above" reported in the first paragraph of this sub-title should be revised as "The following".

Author Response

请参阅附件
